# Structural insights into Cir-mediated killing by the antimicrobial protein Microcin V
Stavros A. Maurakis [1], Angela C. O'Donnell[2], Istvan Botos[1], Rodolfo Ghirlando [1], Bryan W. Davies [2] ✉ & Susan K. Buchanan [1] ✉

Drug-resistant bacteria are a global concern. Novel treatments are needed, but are difficult to develop for Gram-negative species due to the need to traverse the outer membrane to reach targets beneath. A promising solution is found in natural antibiotics which bind outer membrane receptors and co-opt them for import. Exploring this mechanism may open avenues for antibiotic development. An underappreciated class of natural antibiotics are microcins - small antimicrobial proteins secreted by certain bacteria during inter-species competition. Microcins bind outer-membrane receptors of prey species for passage into the periplasm. They have potent activity, bind specific targets, and can control pathobiont expansion and colonization. One microcin, MccV, utilizes the *E. coli* colicin Ia receptor, Cir, for import. Here, we report the first high-resolution structure of the Cir/MccV complex by Cryo-EM, revealing an interaction centered on an electropositive cavity within the Cir extracellular loops. We also report the affinity of MccV for Cir. Lastly, we mutagenized interacting residues and identified key contacts critical to MccV binding, import, and bacteriolysis. Future efforts may help disentangle the mechanisms of microcin killing and will assess relationships between other microcins and their targets to better understand the potential for microcins to be used as antibacterial drugs.

Increasing incidence of drug-resistant bacterial infections represents a growing threat to global health[1,2]. Notably among these, Gram-negative species make up four of the six ESKAPE pathogens[3], and while not listed as an original member of these six, the ubiquitous Gram-negative *Escherichia coli* has also demonstrated a marked increase in antimicrobial drug resistance (AMR) among certain isolates[4]. To be effective, drugs must either traverse the Gram-negative outer membrane to access targets inside or target the membrane directly; both approaches pose obstacles to drug development or clinical deployment[5–8]. Overcoming the barrier to drug uptake would represent an important step forward in drug development. One approach is to take advantage of the innate outer-membrane transport systems already present in the bacteria, such as the Ton system[9,10]. Indeed, antibiotics which bind bacterial siderophores, common substrates of TonB-dependent transporters (TBDTs), and co-opt them for import have been described[11–14]. While most of the efforts in this arena have been limited to siderophore conjugates, an understudied class of bacteriocins called microcins represents another potential target.

Microcins are small (<10 kDa) antibacterial proteins which, when secreted by producer bacteria, localize to and bind specific receptors on the outer-membrane of other bacterial targets, including TBDTs[15–17] (Fig. 1A, B). Once imported, microcins block key functions in the target cell such as inhibiting protein synthesis, inducing double-stranded DNA breakage, or impairing ATP synthesis. Others kill more directly by forming pores in the bacterial membrane[15,18,19]. Harm to producer strains is prevented by production of an immunity protein which localizes to the inner membrane and is thought to act similarly to the cognate immunity proteins for Colicins E1, A, Ia, and Ib[20–24]. Broadly, the microcins are divided into class I and class II; class I members are highly post-translationally modified, while class II are largely unmodified except for the permissible presence of disulfide bonds (class IIa) or C-terminal attachment of iron siderophores (IIb)[25–27]. Besides their promising import phenotypes, the microcins demonstrate several other characteristics which suggest they may have strong applications in combating AMR: they are highly potent, are tolerant of a large pH range, and do not show toxicity to human cells[28–30].

The class IIa MccV (Microcin V, formerly Colicin V), produced mainly by *E. coli* but found in certain other *Enterobacteriaceae*, is well characterized among the microcins[21,31,32]. Once exported, MccV exhibits bactericidal activity against various *Escherichia, Salmonella, Shigella*, and *Klebsiella* species[33]. MccV

[1]Laboratory of Molecular Biology, National Institute of Diabetes and Digestive and Kidney Diseases, National Institutes of Health, Bethesda, MD, USA. [2]Department of Molecular Biosciences, University of Texas at Austin, Austin, TX, USA. ✉e-mail: bwdavies@austin.utexas.edu; susan.buchanan2@nih.gov

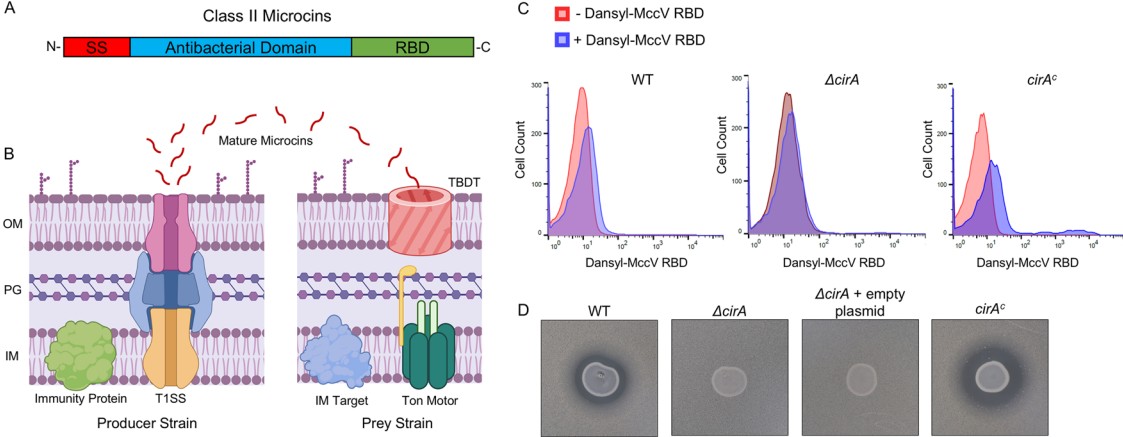

**Fig. 1 | Class II microcins interact with TonB-dependent transporters.**
**A** Schematic representation of Class II microcin domains. SS signal sequence, RBD receptor binding domain. **B** Schematic overview of class II microcin function. Immature pre-microcins are produced in the cytoplasm of producer strains and subsequently exported by Type I secretion (T1SS). This process is accomplished by a C39 peptidase-containing ATP-binding cassette transporter (CvaB, orange), a membrane fusion protein (CvaA, blue), and the outer-membrane efflux pump TolC (magenta). During secretion, CvaB cleaves the signal peptide of pre-microcins, releasing the mature protein for export. Mature microcins in supernatant then localize to the surface of prey bacteria and are imported by binding and exploiting receptors from the Ton (and Tol) system, which harness the proton motive force via the inner membrane Ton motor to energize transport across the outer membrane. Once they have entered the periplasm, microcins are free to interact with their cognate targets at the inner membrane or cytoplasm. Producer strains are protected from auto-bactericidal activity via production of an immunity protein. OM outer membrane, PG peptidoglycan, IM inner membrane, TDBT TonB-dependent transporter. **C** Flow cytometry histograms showing dansyl fluorescence shifts. WT *E. coli* cells treated with dansyl-MccV RBD (blue peaks) exhibit a shift relative to the untreated population (red peaks), and a greater shift is observed when Cir is over-expressed from a plasmid (right graph). This shift is lost when the Cir receptors are absent in a *cirA*::kan mutant (middle graph). **D** Zone of inhibition assays demonstrating the sensitivity of *E. coli* to spotted predator strain secreting WT MccV. The WT *E. coli* lawn demonstrates clearance, indicating sensitivity to MccV. This activity is lost when *cirA* is knocked out. Activity is restored when a plasmid expressing Cir is used to transform the *cirA*::kan strain, while presence of an empty plasmid lacking a complementary *cirA* gene remains resistant.

is recognized by Cir, an iron-regulated TonB-dependent siderophore receptor that is also the target of the antimicrobial Colicin Ia[34–36]. Upon Cir-dependent import to the periplasm, MccV targets the inner-membrane serine transporter SdaC, ultimately resulting in collapse of membrane potential due to membrane pore formation[37,38]. However, neither the specific nature of MccV recognition of Cir, nor its interactions with the Ton motor (TonB, ExbB, ExbD), have previously been described.

To understand how MccV binds Cir, we solved the first structure of this complex via cryogenic electron microscopy (cryo-EM). Combined with binding assays utilizing purified biomolecules, we report that Cir and MccV bind with affinity in the nanomolar range, and that the structural nature of the binding interaction is conserved between known Cir ligands. Lastly, using site-directed mutagenesis, we interrogated the Cir/MccV binding and import mechanism and identified critical amino acid residues responsible for binding and effective import and bactericidal activity by MccV. Ultimately, these findings provide a critical advancement in the understanding of microcin biology and represent an important step toward characterizing the microcins as potential treatments against high-priority AMR bacteria.

## Results

### MccV binds and kills Cir-expressing *E. coli*
Previous reports indicate that MccV is recognized by Cir, and the interaction results in bacterial death from disruption of membrane potential and an as-yet-undescribed interaction with the inner-membrane serine transporter SdaC[34,37,38]. However, little is known about the exact nature of their binding interaction or the structural nature of MccV docking with Cir. As such, we sought to interrogate these topics to gain further insights into microcin form and function. Previous research demonstrated that the 32 C-terminal amino acids of MccV were sufficient to confer specificity to Cir for microcin cell entry[39]. To validate this genetic finding, we first established whether the C-terminal receptor binding domain (RBD) of MccV alone was capable of binding Cir-expressing *E. coli*. To this end, we synthesized a dansyl-tagged peptide comprised of the 32 RBD residues of MccV and tested for binding using flow cytometry (Fig. 1C). This peptide caused a fluorescence shift in a wildtype culture of *E. coli* but not in the isogenic *cirA* deletion strain. This

effect was complemented by the expression of *cirA* from a plasmid. From here, we set out to define a bactericidal assay which demonstrated Cir-dependent bacteriolysis by secreted MccV. We demonstrated that when *E. coli* producing both MccV and its immunity protein are spotted on a lawn of sensitive *E. coli*, a clear zone of inhibition is observed, indicating MccV antibacterial action against the sensitive strain. This zone was lost when *cirA* was deleted, while sensitivity was restored by *cirA* expression in trans. (Fig. 1D). This killing assay gave us a framework within which to probe phenotypic changes as we further characterized the interaction between the proteins.

### Structure determination of the Cir/MccV complex
4-Met substituted *E. coli* Cir[35] was expressed into native membranes of *E. coli* strain BL21 (DE3), which were then isolated. The membrane fraction was incubated with purified MccV and then solubilized using n-dodecyl-β-D-maltoside (DDM) and purified using a 10X N-terminal histidine tag on Cir. After affinity chromatography, DDM was replaced with nonionic amphipol (NAPol)[40] as a stabilizing agent for Cir, and the remaining purification steps were performed in the absence of detergent. The NAPol-stabilized Cir/MccV complex was then run over size exclusion chromatography. Under these conditions, Cir and MccV co-eluted cleanly as a single peak. Cryo-EM analysis of these fractions showed homogenous, well-distributed particles, which were used for data collection and single particle analysis. (Supplementary Fig. 1).

In total, 102,419 particles were used to calculate the three-dimensional (3D) map of the Cir/MccV complex in NAPol at 2.9 Å resolution (Fig. 2, Supplementary Fig. 2 and Table 1). A predicted model was generated using AlphaFold2[41] and fit into the density map, then manually adjusted for best fit (Supplementary Fig. 3). The entire structure was then refined to a final map-to-model correlation coefficient of 0.88. Our cryo-EM structure of Cir shows the prototypical architecture of a TBDT, including a 22-stranded β-barrel occluded by an N-terminal plug, and 11 extracellular loops of variable length[42]. More specifically, it has overall dimensions consistent with those previously observed during crystallographic analysis of the Cir apo-form (PDB code 2HDF): an ellipsoid β-barrel approximately 40 Å high with

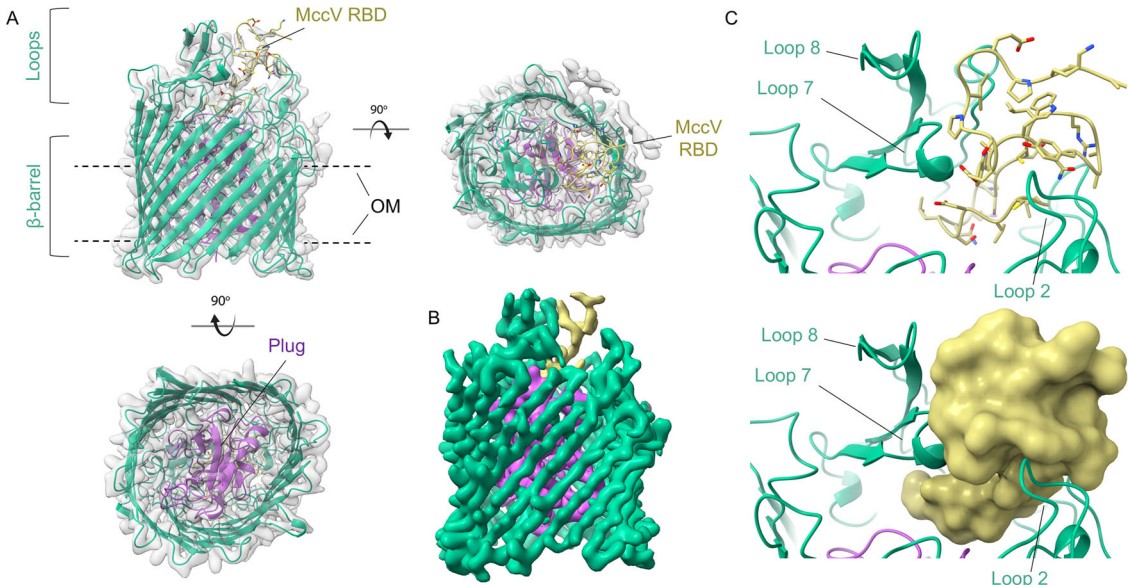

**Fig. 2 | Structure of Cir bound by the MccV receptor binding domain. A** Cryo-EM structure of the Cir/MccV RBD complex. The β-barrel and extracellular loops of Cir are shown as green ribbon with the plug domain highlighted in purple. The MccV RBD is shown in gold. The structure is inset into the cryo-EM density map with the NAPol micelle subtracted. OM outer membrane. **B** Cir/MccV RBD cryo-EM map colored to match model (contour level 0.120). **C** Illustration of the Cir/MccV binding pocket shown as cartoon (top) and as MccV surface (bottom). MccV interacts with the Cir extracellular loops in an area flanked by loops 2, 7, and 8 at the top and the plug domain below. This map and accompanying coordinates have been deposited to EMDB and PDB under codes EMD-49565 and 9NN6, respectively.

inner dimensions of approximately 40 Å across the short axis and 50 Å across the long axis, and extracellular loops 7 and 8 being the longest, occupying most of the area across the top of the barrel[35] (Fig. 2A, B). Loop 2, made up of residues 210–231, was not resolved in the original crystal structure but was visible in the cryo-EM map. This loop is the third largest and sits on the opposite side of the β-barrel from loops 7 and 8.

While mature MccV consists of residues alanine 16 to leucine 103 (residues 1–88 by our numbering), the cryo-EM map showed strong signal only for residues 57–88, which correspond to the C-terminal RBD (Fig. 2B). Inspection of this section of the map showed that the MccV RBD settles into a binding pocket atop the plug domain and flanked on one side by Cir loops 7 and 8, and on the other by loop 2 (Fig. 2C). In total, an interacting surface of 1232.8 Å² is buried during complex formation, representing 44.7% of the total MccV surface found in the model. Strikingly, this is the same space occupied by the Colicin Ia R-domain upon its docking with Cir[35].

## MccV binds Cir similarly to Colicin Ia

Upon observing that MccV bound Cir in the same position as Colicin Ia, we wondered whether there were similarities in their binding modes and mechanisms as well. When compared to apo-Cir, we observed a large deflection away from the top of the plug by loops 7 and 8, with the tip of each loop moving 11.5 Å and 12 Å, respectively, from its position in the apo-form (Fig. 3A and Movie 1). This change in conformation reveals the binding pocket and opens a space for MccV to occupy, which would otherwise be occluded (Fig. 3B). Parts of Cir loop 2 also show close proximity to MccV, with sidechains from each protein separated by as little as 3.5 Å. Comparison of the apo- and ligand-bound structures shows an outward twisting movement at the base of this loop, suggesting that it too may be displaced during binding, but because this loop was not well resolved in the apo-form crystal structure, we could not assess this fully. We also observed slight movement in the residues corresponding to the Cir TonB Box[43] (T32-A37, Movie 2), suggesting a possible adaptation of a plug conformation which allows access and binding by TonB. This is consistent with previous reports which showed the necessity of TonB for MccV-dependent bactericidal activity[34].

The rearrangement described above mimics the action of loops 7 and 8 upon Colicin Ia binding, which results in an even greater displacement: 17 Å

in loop 7 and 21 Å in loop 8. Because MccV occupies the same space and causes approximately the same conformational changes to Cir as does Colicin Ia, we hypothesized that the nature of the binding interactions may be similar. The H-bonding network between Colicin Ia and Cir is well understood. Of note, R436 of Cir was shown to interact with D350 of Colicin Ia, with Colicin E369 also in proximity. Likewise, Colicin residues D358 and E357 are in proximity to Cir R490 and R116, respectively[35]. We noted Cir R116, R436, and R490 in our own model and probed the interaction of these side chains with MccV. Strikingly, these residues appear to form an electropositive cavity at the base of the loop 7/loop8 area above the plug, with the sidechain of MccV D85 settled into the middle (Fig. 4A, B, Supplementary Fig. 4A, B). When in complex, charged atoms from D85 are located 3 Å, 4.3 Å, and 2.8 Å from complementary charges on R116, R436, and R490, respectively (Fig. 4C, D). We also note a 2.8 Å salt bridge formed between OD1 from D85 and NH2 from R490. We further confirmed that MccV occupies the same binding pocket as Colicin Ia by performing competitive binding experiments. We fixed His-tagged Cir either alone or in complex with the R Domain of Colicin Ia onto nickel-coated ELISA plates and probed with MccV. When the Colicin Ia R Domain was already bound to Cir, the amount of MccV binding detected was statistically indistinguishable from what was observed when no Cir was present on the plate, and bound significantly less than it did for apo-Cir (Supplementary Fig. 4C, D). To further probe the Cir/MccV interaction, we performed clustering analysis using PICKLUSTER[44]. This analysis reinforced our supposition that the arginine cavity coordinated MccV binding, suggesting an interaction network anchored by R436 and R490, with contributions from C435 and S464. An additional minor interface was predicted between Cir R513 and MccV E61. With putative interacting residues identified, we set out to validate these findings via functional and binding analyses.

## Mutagenesis impairs MccV binding, import, and bactericidal function

We generated a series of mutations in Cir and MccV to probe their effects on MccV binding and function, with a primary focus on alanine substitution and swapping charged residues to the opposite charge. Cir mutants were stable and consistent with the wildtype when expressed in *E. coli* and mutations in both proteins did not affect purification (Supplementary

**Table 1 | Cryo-EM data collection and structure statistics**

| | Cir/MccV |
|---|---|
| **Data collection** | |
| Magnification | 105,000× |
| Voltage (keV) | 300 |
| Total Dose (e/Å²) | 59.65 |
| Number of frames | 30 |
| Exposure time (s/frame) | 0.075 |
| Defocus range (μm) | −0.8 to −2.4 |
| Pixel size (Å) | 0.83 |
| **Image processing** | |
| Movies collected | 7606 |
| Micrographs selected | 5960 |
| Particles extracted | 7,726,727 |
| Final map particles | 102,419 |
| Symmetry imposed | C1 |
| FSC threshold | 0.143 |
| Final map resolution (Å) | 2.9 |
| Resolution range (Å) | 3.06–2.71 |
| **Atomic model** | |
| Protein residues | 664 |
| Chains | 2 |
| **Validation** | |
| Ramachandran favored (%) | 97.73 |
| Ramachandran allowed (%) | 2.27 |
| Ramachandran outliers (%) | 0 |
| Rotamer outliers (%) | 0 |
| RMSD bond lengths (Å) (# > 4σ) | 0.002 (0) |
| RMSD bond angles (°) (# > 4σ) | 0.419 (0) |
| Clash score | 5.60 |
| Map CC (mask) | 0.87 |
| Map CC (volume) | 0.88 |
| **Deposition IDs** | |
| PBD | 9NN6 |
| EMDB | EMD-49565 |

Fig. 5). We first assessed whether the mutations impacted Cir sensitivity to MccV killing. To this end, we performed zone of inhibition assays comparing MccV sensitivity between WT and *cirA* mutant constructs. We observed a range of loss of sensitivity in the C435A, R436A/E, R490A/E, and R436E/R490E Cir variants, as well as in D85A and D85R MccV (Fig. 5A). We hypothesized that the impaired MccV function was concomitant with diminished or abolished binding, so we next set out to evaluate potential binding deficiencies introduced by our mutations. We first calculated the affinity between the wildtype proteins using biolayer interferometry (BLI). Using purified, biotinylated MccV, we loaded streptavidin coated BLI sensors which were then probed with dilutions of Cir, resulting in a dose-dependent response trace (Fig. 5B). When fitted to a single site binding model, the rate constants corresponding to the first 30 s of association and dissociation suggested a high affinity with $K_D = 155 \pm 30$ nM. We further calculated a steady state equilibrium affinity using a similar 1:1 binding model, which returned a calculated $K_D$ of $1.7 \pm 1.2$ μM, but may be slightly less accurate due to incomplete equilibrium during association phase. Because this is the first time reporting the affinity of these partners, and because we planned to deploy this BLI-based approach for further characterization, we elected to validate our initial findings using another independent method. For this, we utilized sedimentation velocity analytical ultracentrifugation (SV-AUC). SV-AUC analysis of NAPol-stabilized Cir and its mixtures with MccV resulted in a single-site model $K_D$ of $0.6 \pm 0.2$ μM, which is within range of our BLI results (Supplementary Fig. 6). Based on concurrence of two experimental approaches, we confidently report a 1:1 binding stoichiometry and a $K_D$ in the mid-nanomolar to low-micromolar range. With a baseline binding affinity established, we purified both MccV mutants and the Cir variants with altered MccV sensitivity and performed the same BLI analysis for each (Fig. 5C, D). D85A and D85R showed similar reductions in Cir binding relative to WT, with approximately a 40–45% reduction in response shift for the same probe concentration of 5 μM. There was more variation from the Cir mutations. C435A showed the smallest change and reached a peak response shift identical to that of the WT, albeit with a slower on-rate. R436E/R490E showed a moderate loss of approximately 55% response shift compared to WT. The most pronounced defect was observed in R490A, in which binding was almost completely abolished and response shift was similar to that observed when no probe was added.

Finally, we considered the role of the Ton motor on MccV import after initial binding to Cir at the cell surface. As mentioned, we observed a slight movement in the Cir residues comprising the TonB Box (Movie 2), and it has previously been shown that inactivation of the *tonB* and *exbB* genes prevented MccV uptake[34]. In addition, TonB Box point mutations which affect its crosslinking pattern with TonB are known to cause TonB uncoupling and subsequent loss of the import phenotye[45]. With this in mind, we utilized a Cir TonB Box mutant (M33C/V35P, Fig. 6A) in both our agar plate killing assay and BLI to confirm that Cir, via the Ton motor, was the primary means of MccV traversal through the outer membrane (Fig. 6B–D). When M33C/V35P-expressing cells were grown in the presence of MccV-secreting cells, we observed no zone of inhibition in the secretion area, indicative of no successful MccV import. Conversely, when we added M33C/V35P to BLI plates with WT MccV, we observed similar $K_D$ values and no reduction in binding signal compared to WT Cir at the same dilution, suggesting initial binding prior to import was unaffected. Collectively, these observations confirm the role of the electropositive cavity in the Cir binding pocket as a key target for MccV binding, and we conclude that Cir, via TonB-dependent import, is also the key translocation path for MccV into the periplasm.

Taken together, our findings in the current study demonstrate the specific interaction between Cir and MccV, built upon a high-affinity binding event reminiscent of that observed in another Cir ligand, Colicin Ia. The flexible extracellular loops of Cir facilitate a conformational change to reveal a normally occluded region of electropositivity, and secreted MccV kills Cir-expressing *E. coli* only when this cavity is intact and when the target cell has a functioning Ton system.

## Discussion

Microcins were first described as eco-active, growth-inhibitory molecules of low molecular weight produced by Gram-negatives; they were shown to resist proteases and extremes of temperature and pH[46,47]. It was during these early characterizations that the possibility of using microcins as targeted antimicrobials was considered, and while still a promising candidate, they are as-yet undeveloped and understudied[48,49]. Few microcins have been characterized to the point of having their functions confirmed, and most available information is restricted to the genetic or bioinformatic level, with the exception of Microcin D93 for which some structural data are available[50]. For MccV, the genes for microcin production and export are known and have served as the blueprint for identifying other Class II species[21,22,32].

TBDTs have long been a focal point in understanding active transport across Gram-negative membranes, with the first examples of their crystal structures being solved in the nineteen nineties[51,52]. They play critical roles in bacterial metabolic homeostasis by facilitating the uptake of nutrients including metal ions, vitamins, and carbohydrates[9,42]. Cir is an iron regulated TBDT found in *E. coli* which coordinates iron uptake by binding

**Fig. 3 | Cir undergoes conformational changes to facilitate MccV docking. A** Cir/MccV RBD (MccV RBD not shown) superimposed onto apo-Cir (PDB code 2HDF, blue). Loops 7 and 8 from the complex structure are displaced relative to their positions in the ligand-free structure by 11.5 Å and 12 Å, respectively. **B** MccV overlaps the space vacated by loops 7 and 8. Left: apo-Cir with MccV RBD (cartoon inset into surface, gold) superimposed. MccV occupies the same space as loops 7 and 8 from the apo structure. Right: The complex structure with loops 7 and 8 deflected to facilitate MccV docking.

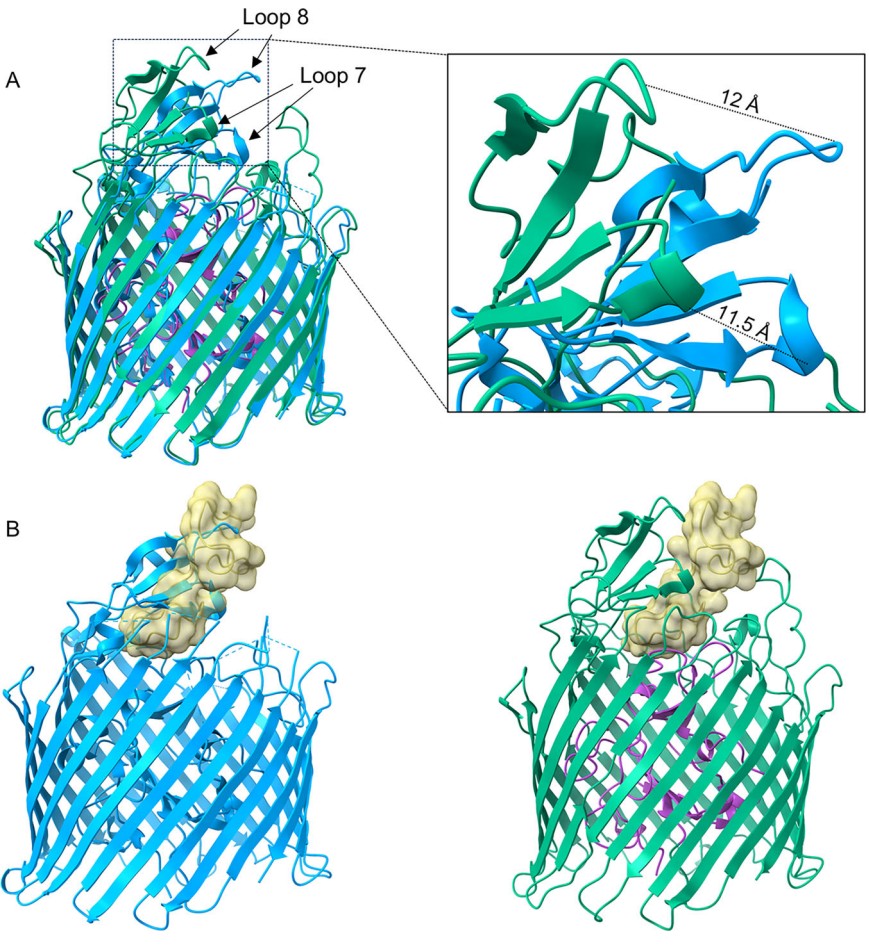

catecholate siderophores such as dihydroxybenzoyl serine[53,54], but can also be co-opted by bacteriocins during interspecies competition. The high-resolution structure of MccV bound to Cir allows us to begin detangling the molecular nature of MccV binding to target cells, and provides a critical step in evaluating microcin potential in structure-based drug design.

Targeting TBDTs for drug and vaccine development is a topic which has gained momentum over the past two decades. Recently, the β-lactam cefiderocol, which uses siderophore-like properties to access the periplasm directly via TonB-dependent transport, has shown promising efficacy against multidrug-resistant isolates, including the ESKAPE pathogen *Acinetobacter baumannii*[55–57]. Earlier such efforts utilized siderophore conjugation to the monobactam aztreonam, examples of which reached phase I trials[58,59]. The TBDTs themselves exhibit many features which make them promising targets for either drugs or vaccines: they are surface exposed; they show wide distribution in pathogenic strains; they are critical and in some cases strictly required for establishing infections[60–63]; and for vaccines specifically, can elicit protective immune responses[64,65]. Moreover, their domains and function are generally well-conserved.

MccV binds Cir atop the plug domain in a pocket normally occluded by loops 7 and 8, which hinge open to facilitate ligand entry. This is consistent with previous reports which demonstrate substrate-dependent rearrangement of extracellular loops in other TBDTs[66,67]. For Cir, an electropositive cavity revealed by this rearrangement, flanked by residues R116, R436, and R490, is key in coordinating interaction with D85 from MccV, a binding condition that is shared with Cir ligand Colicin Ia. We submitted the Cir amino acid sequence to ConSurf[68] (https://consurf.tau.ac.il) to assess whether these residues were conserved over *E. coli* isolates (Supplementary Fig. 7). R116, which sits in the highly-conserved plug domain, is maximally conserved across queries, while R436 and the adjoining C435 show moderate conservation. R490 however shows moderate variability, which one

could hypothesize is the result of evolutionary pressure from bacteriocins in the intestinal chemosphere, especially considering our data which showed it to be the residue with the most profound impact on MccV binding. This prompts interesting questions regarding the evolutionary relationships between microcins and their receptors, which future studies may aim to clarify.

Mutation of residues at the Cir/MccV binding pocket resulted in clear deficiencies in bacteriolysis and binding, but it is notable that binding was only altogether eliminated in the R490A Cir variant. Similarly, while D85A MccV produced a zone of inhibition significantly smaller than seen for WT MccV, a minor zone was still visible which was mostly absent from D85R. This raises interesting questions about the relationship between affinity and activity. R490A had lower MccV affinity than R436E/R490E despite sharing a mutated residue, but both saw the same loss of MccV sensitivity. Conversely, the affinity of C435A was within the margin of error of D85R and nearly within for D85A, but each showed slightly different phenotypes for sensitivity. Unfortunately, because our killing assay relies on natural export of MccV from producer strains instead of the addition of specific amounts of protein, we were not able to determine a true minimum inhibitory concentration for each mutant. One could speculate that single point mutations in MccV are sufficient to markedly reduce binding and import efficiency, and in doing so reduce but not altogether eliminate killing, whereas changes in the binding pocket of Cir may have architectural/structural effects which preclude import altogether. Indeed, a not-perfectly-correlative relationship between affinity and activity is consistent with our observation that D85A had slightly lower affinity than D85R, but showed less loss of sensitivity, suggesting a complex relationship between binding, import, and activity. Considering these observations together, one must be mindful when discussing MccV binding and allow that while the charged cavity is a central player in initial recognition, the collective network of interacting residues is

**Fig. 4 | MccV interacts with an electropositive pocket in the Cir loops.** Three arginine residues (Arg 436 from loop 7, Arg 490 from loop 8, and Arg 116 from the plug) form an electropositive cavity within the Cir ligand-binding pocket. When in complex, the sidechain of aspartate 85 from MccV fills the space inside the cavity, creating a putative electrostatic interaction. **A**, **B** A shot-reverse shot illustration of the interacting residues shown as cartoon (top) and with electrostatic surface colored (bottom). Arginine cavity electropositive surface is shown on the left of panel B, with the electronegative Asp 85 surface shown on the right. **C**, **D** Asp 85 settles in close proximity to the Cir aspartates. Atomic distances between the relevant atoms of Asp 85 and each of the arginines were determined and are illustrated in (**C**), with sidechain heteroatom surfaces shown in (**D**). Interactions shown: R116 NH1 with D85 OD2, R436 NE with D85 OD2, and R490 NH2 with D85 OD1.

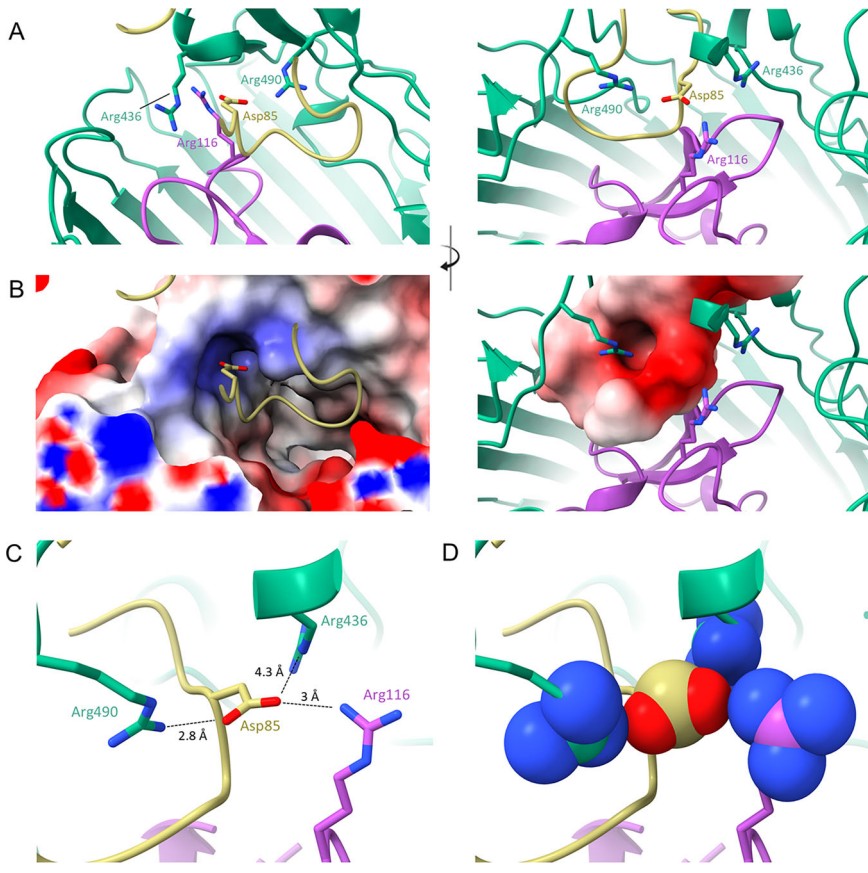

likely to be much larger. Similarly, an allowance must be made for the fact that our understanding of the full mechanism of MccV import remains limited. Our data are consistent with previous reports suggesting the necessity of a functional TonB[34], an observation which is also shared for Colicin Ia[35], but we as yet do not have a clear picture of the events which follow MccV binding at the Cir loops or whether it interacts with other targets prior to locating SdaC. This remains an interesting area of investigation, prompting questions over the requirement, for example, for MccV unfolding to traverse the Cir pore or whether other proteins are recruited during the process. It is especially interesting that MccV and Colicin Ia share a transporter in light of their considerable size difference[69] (Supplementary Fig. 8), and considering that MccV is not predicted to have a TonB box of its own as one would find in the Group B colicins like Ia. Future efforts in this area are expected to investigate this.

A final consideration involves the use of microcins as antimicrobials and how the microcins, when deployed, will interact with the host niche inhabited by their cognate pathogen. Specifically, one wonders about competition for protein binding sites between the microcin and the "intended" substrate of its target, be it a catecholate siderophore as for Cir, or otherwise. We showed that MccV binds Cir with an affinity of approximately 500 nM, but other siderophore-receptor interactions have been shown previously to be much tighter[70]. In direct competition, then, the microcins may be naturally disadvantaged by the presence of other substrates. Further studies between known microcin/receptor pairs, including MccV and Cir, should aim to assess this obstacle and consider potential protein engineering strategies that may be necessary to maximize microcin efficacy.

## Methods

### Bacterial strains and plasmids
All bacterial strains, primer sequences, and plasmid maps used in this study are available upon request. The mature coding region of *E. coli cirA* (residues W338, L343, F589, and V591 in this plasmid had previously been mutated to methionine during crystallization experiments with this protein, so the 4-Met designation has been retained in this report) was cloned into a modified pET-9 vector (Novagen) containing a *pelB* signal sequence followed by a 10× histidine tag and a tobacco etch virus (TEV) protease recognition site[71] upstream of *cirA*. Cir mutants were generated in this plasmid via Q5 site-directed mutagenesis (New England Biolabs). For MccV, the mature coding region was cloned into pET-11M (Novagen) to generate a maltose binding protein (MBP) fusion. The final construct carried MBP bearing an N-terminal 6X histidine tag, followed by a TEV cleavage site and then MccV. Mutant MccV was generated from this plasmid using Q5. All plasmids were confirmed by sequencing (Psomagen).

### Protein expression and purification
Cir (WT and mutants) was expressed in BL21 (DE3) *E. coli* (New England Biolabs) as follows. Chemically competent cells were transformed with appropriate plasmids and selected on Luria-Bertani (LB) medium containing 50 μg/mL kanamycin. A single colony was selected and confirmed via western blot to produce his-tagged Cir, then stocked in 25% glycerol. For large scale expression, a 5 mL starter culture from stock was grown in LB + antibiotics at 37 °C overnight. This starter was then used to inoculate 2 flasks containing 1 L of terrific broth (TB) + antibiotics, and these were grown at 20 °C with 225 RPM shaking and no inducer added, instead allowing the promoter to slowly leak. Cell pellets were collected by centrifugation (10,000 × *g*, 10 min, 4 °C) and resuspended in lysis buffer (50 mM Tris-HCl pH 7.5, 200 mM NaCl, 10 mM MgCl₂, 10 μg/mL DNase I (Goldbio), 100 μg/mL 4-(2-aminoethyl)benzenesulfonyl fluoride hydrochloride [AEBSF, GoldBio]) at a ratio of 100 mL buffer for every 25 g of cell pellet.

Once fully resuspended, cells were lysed via two passages through an Emulsiflex C3 (Avestin) with a homogenizing pressure of ~17,500 PSI. The lysate was then clarified by centrifugation (20,000 × *g*, 10 min, 4 °C) and the

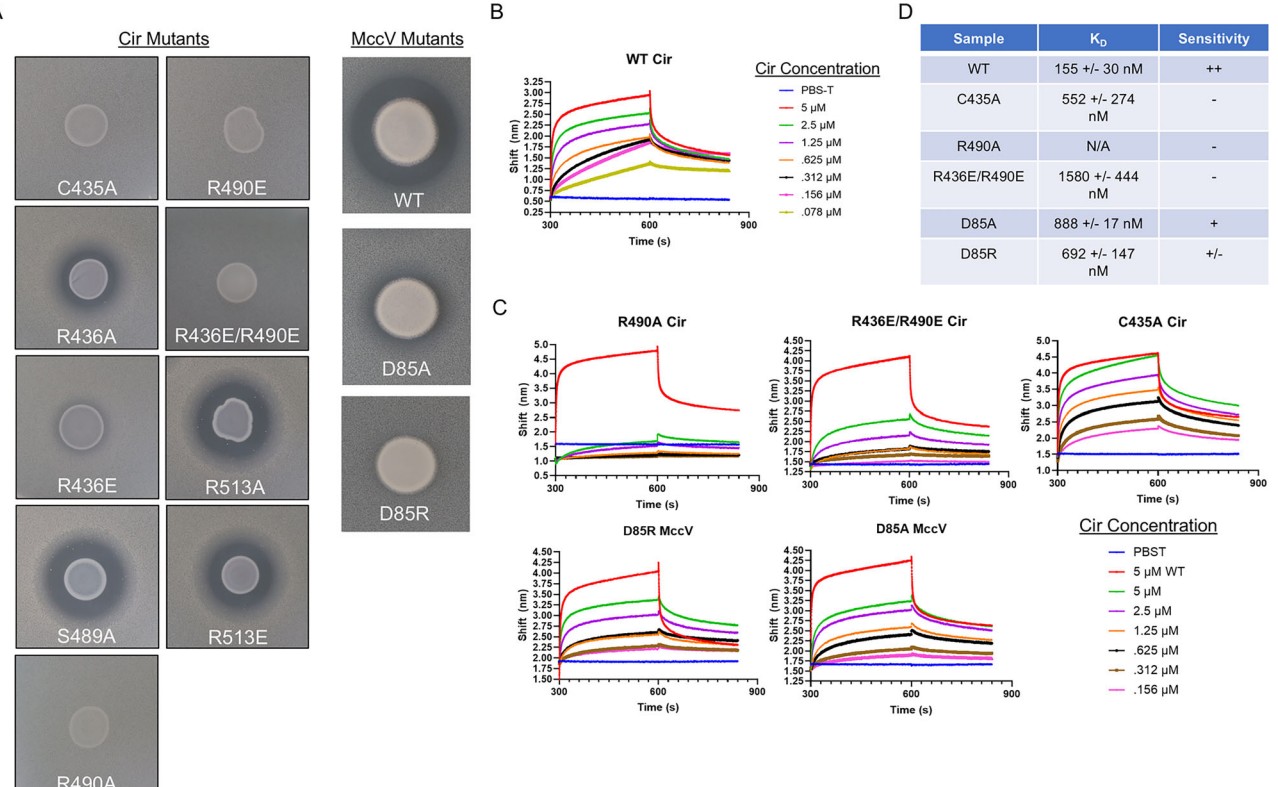

**Fig. 5 | Binding pocket mutants show decreased bactericidal activity and binding.**
**A** Zone of inhibition assays demonstrating sensitivity of Cir mutants to WT MccV. Cir mutations C435A, R436E, R490E, R490A, and R436E/R490E exhibit losses in sensitivity to WT MccV secreted from *E. coli*. The third column shows that the toxic activity of MccV is attenuated when residue D85 is mutated. **B** MccV binds Cir with high affinity. Purified MccV was biotinylated and loaded onto streptavidin-coated biolayer interferometry (BLI) sensors at a concentration of 100 nM, then probed with a dilution series (5 μM ->78 nM in twofold steps) of wildtype (WT) Cir to

determine affinity. **C** Mutant binding assays. 100 nM WT or mutant biotinylated MccV was loaded onto BLI sensors and probed with dilutions of WT or mutant Cir to compare binding against the WT partners. In each mutant assay, 5 μM WT Cir or 100 nM WT MccV was included as a positive control (red trace). For all BLI experiments, buffer alone (PBS-T) was added as a negative control. Representative BLI traces are shown from *n* = 3 experiments. **D** Table showing affinity and Cir/ MccV-dependent sensitivity for mutants with relevant phenotypes. Note that R490A failed to properly fit to the binding model due to poor dose response.

supernatant collected. Lysate was mixed with 1% Triton X-100 and gently stirred for 30 min at room temperature. Membrane pellets were collected by centrifugation in a Beckman Type 45 Ti rotor (234,000 × *g*, 45 min, 4 °C) and supernatant was discarded. Membranes were resuspended in 1X phosphate buffered saline (PBS) using a Dounce homogenizer. This solution was mixed with an excess volume of purified MccV (described below) for 2 h, then final concentrations of 40 mM imidazole and 1% DDM were added. This mixture was gently mixed for 2 h at room temperature to solubilize the membranes. After solubilization, the solution was clarified by centrifugation in a Beckman Type 70 Ti rotor (310,000 × *g*, 45 min, 4 °C) and the soluble fraction was collected. This was filtered through a 0.22 μm filter then applied to an AKTA purifier (GE) for nickel affinity chromatography. The sample was looped ten times over a 5 mL HisTrap column (Cytiva) that was equilibrated with PBS pH 7.5 plus 0.1% DDM prior to sample application. His-tagged Cir was eluted by stepwise addition of imidazole in the same buffer, with the majority eluting at 300 mM imidazole.

Fractions were analyzed by SDS-PAGE and the cleanest were carried forward. These fractions were pooled and mixed with a threefold mass excess of NAPol (Anatrace) and gently rocked at room temperature for roughly 1 h. DDM was removed by addition of freshly-hydrated Bio-Beads SM-2 (Bio-Rad) and gentle rocking for 1 h. The beads were removed by filtering through a 0.22 μm filter and the resulting protein solution was concentrated to a final volume of 500 μL using a 50 kDa cutoff centrifugal filter (Millipore). This sample was applied to a Superdex 200 Increase 10/300 GL column (Cytiva) equilibrated with 1× PBS for size exclusion chromatography. Peak fractions were again analyzed by SDS-PAGE, and sample

homogeneity from the cleanest fractions was further confirmed by mass photometry[72] using a OneMP mass photometer (Refeyn).

MccV (WT and mutant) was expressed in BL21 (DE3) *E. coli* as follows. Chemically competent cells were transformed and selected on LB + 50 μg/mL Kanamycin, then stocked as described above. A 5 mL starter culture was grown overnight as described and used to inoculate a flask containing 1 L 2xYT medium + antibiotics. This culture was grown at 37 °C at 225 RPM shaking until the it reached an $OD_{600}$ of 0.6–0.8, at which point protein expression was induced by addition of 1 mM Isopropyl β-D-1-thiogalactopyranoside (IPTG). Protein expression continued for 3 h before cell pellets were collected by centrifugation and resuspended in lysis buffer as described above. Cells were lysed via 5 min of looping through the Emulsiflex C3 and then clarified by centrifugation. The lysate was applied to an AKTA purifier and loaded onto a 5 mL HisTrap column in 1× PBS to capture the His-tagged MBP fusion partner. MBP_MccV eluted at a range from 50 to 300 mM imidazole in 1× PBS. Fractions were subjected to SDS-PAGE and relevant fractions were pooled and mixed with 2 mM dithiothreitol (DTT) and 2 mg of His-tagged TEV protease which had been purified as described elsewhere[73]. Briefly, TEV was expressed from Rosetta *E. coli* in autoinduction medium, after which cells were lysed by passage through the Emulsiflex C3 then filtered through a 0.22 μm pore vacuum unit. The sample was loaded first onto a 15 mL nickel column on an AKTA purifier and eluted in 50 mM $K_2HPO_4$, pH 7.5, 200 mM NaCl, 10% glycerol, and 500 mM imidazole. Relevant fractions were pooled and diluted with 15 volumes of 10 mM Tris, pH 7.5, 0.3 mM TCEP and loaded onto a 10 mL SP Sepharose column. TEV was eluted by linear gradient addition of 1 M NaCl in the same Tris buffer. For MBP_MccV cleavage, the protein/TEV mixture

**Fig. 6 | Mutation of the Cir TonB box inhibits MccV killing, not binding. A** A zoom-in of the TonB box within the N-Terminal domain of Cir, highlighted in orange and with the mutated residues labeled. **B** Zone of inhibition assays using WT MccV secreted onto a lawn of WT or M33C/V35P Cir-expressing cells. We observed a zone of inhibition only when WT Cir was produced. **C** Comparison table for affinity and sensitivity for WT and M33C/V35P Cir. **D** BLI assay using WT and M33C/V35P Cir on WT MccV. 100 nM MccV was loaded to BLI sensors and they were subsequently probed with dilutions of M33C/V35P Cir, with 5 μM WT Cir included as a positive control and buffer only as a negative. Trace is representative of *n* = 4 experiments from two independent protein preparations.

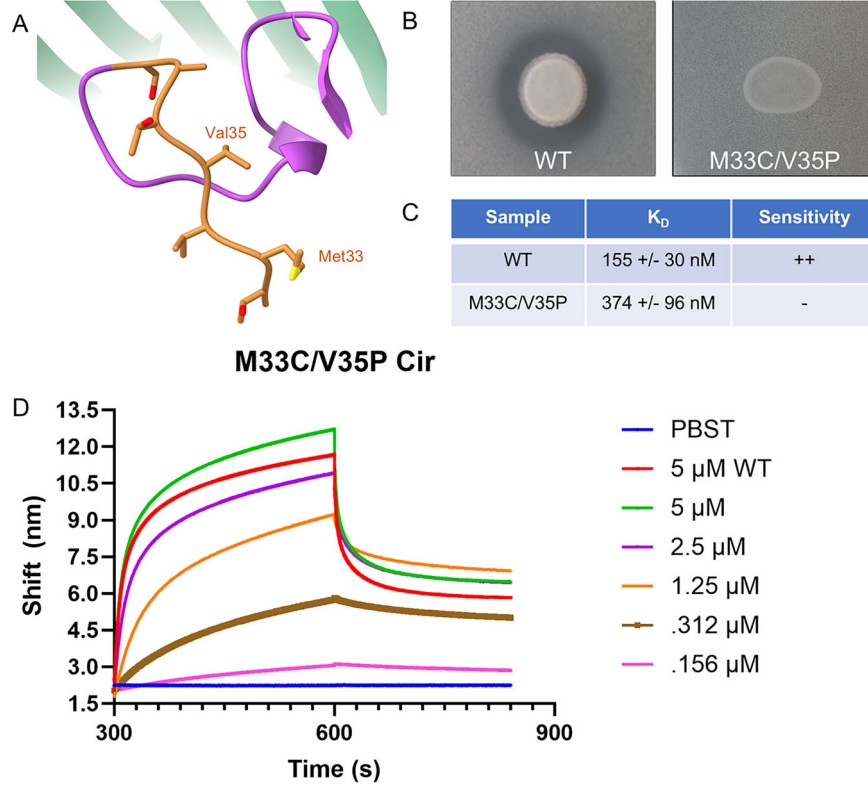

| Sample | K_D | Sensitivity |
|---|---|---|
| WT | 155 +/- 30 nM | ++ |
| M33C/V35P | 374 +/- 96 nM | - |

was pipetted into pre-hydrated 3 kDa cutoff dialysis tubing (Millipore) and dialyzed against an excess of 1× PBS for 48 h at 4 °C while TEV cleavage of the MBP partner proceeded. After dialysis, the resulting solution was passed again over a nickel column to recapture the his-tagged TEV and MBP, and the flowthrough was collected. The remaining purified, untagged MccV was concentrated in a 3 kDa cutoff centrifugal filter and its purity assessed by SDS-PAGE. For complex formation with Cir, the entire MccV prep was mixed with Cir in native *E. coli* membranes prior to detergent solubilization, resulting in approximately a twofold molar excess of MccV to Cir.

The receptor binding domain (R Domain) of Colicin Ia was expressed as inclusion bodies and refolded in the presence of Cir approximately as previously described[35] with minor modifications. The R Domain was cloned into a pET-17b plasmid and used to transform BL21 (DE3) *E. coli*. An overnight starter culture was grown in 5 mL LB + 100 μg/mL Carbenicillin, and the next day the entire starter was subcultured into 1 L 2xYT medium that had been pre-warmed to 37 °C. Antibiotics were added and this culture was grown at 37 °C and 225 RPM until the OD_{600} reached 0.6. 1 mM IPTG was then added and expression was allowed to proceed for 3 h. The cell pellet was collected via centrifugation as described for the other proteins. The 7 g pellet was resuspended in 30 mL lysis buffer (same as above) and lysed by 5 min of looping through the Emulsiflex C3. Inclusion bodies were collected by centrifugation at 8000 × *g* for 10 min at 4 °C, and supernatant was discarded. Pelleted inclusion bodies were then resuspended in 50 mM Tris pH 8.0, 200 mM NaCl, and 2% Triton X-100 for 30 min at room temperature. The mixture was centrifuged as above and then washed twice more in the same buffer minus Triton X-100. The washed inclusion bodies were then resuspended in 20 mL 8 M urea and centrifuged as before to remove insoluble material. Solubilized material was then slowly dripped into 200 mL of stirring 50 mM Tris pH 8, 200 mM NaCl, 550 mM sucrose for rapid dilution, which was then mixed at room temperature overnight. The next day, the refolding mixture was concentrated to ~10 mL using 10,000 MWCO filters and the protein concentration was determined. This mixture was then added at a four-fold excess to freshly prepared Cir in NAPol and allowed to mix for 30 min at 4 °C. The mixture was again concentrated using a 50,000

MWCO filter and the complex was purified on a Superdex 200 Increase 10/300 GL column in PBS pH 7.5, with peak fractions analyzed by SDS-PAGE.

### Cryo-EM sample preparation and data acquisition

3 μL of Cir/MccV complex (7.8 mg/mL) was applied to a Quantifoil R 1.2/1.3 300 mesh cryo-EM grid (Protochips, Inc.) that had been glow discharged for 45 s at 15 mA prior to sample application. The grid was blotted and plunged immediately into liquid ethane using a Vitrobot Mark IV (ThermoFisher). Blotting conditions were as follows: 4 °C, 100% humidity, blot force +5, blot time 3–7 s, wait time 0 s. EM data were collected using SerialEM[74] on a Titan Krios G3 (ThermoFisher) equipped with an Imaging Filter Quantum LS and a K3 direct electron detector (Gatan) with an acceleration voltage of 300 keV. The calibrated pixel size was 0.415 Å/pixel in super-resolution mode. A total of 7606 dose-fractionated movies were collected with acquisition software operating in counting mode. The total electron dose was 60 e/Å² over 30 frames, and a nominal defocus range of −0.8 to −2.4 μm.

### Data processing and model building

All data processing was performed in CryoSPARC v4[75]. Movies were imported then gain and motion corrected using patch motion correction with a binning factor of two (0.83 Å/pixel). Patch CTF was used to estimate CTF parameters. Motion corrected frames were curated and 5960 exposures were retained based on CTF fit resolution and relative ice thickness. A blob picker was used on 500 exposures and identified ~1.1 million particles which were extracted (bin 2) and submitted to 2D classification. 134,798 particles were selected and used to generate two Ab-Initio classes, while the rejected particles were used to generate four "junk" Ab-Initio classes. The 2-class Ab-Initio job was plugged immediately into heterogeneous refinement and the better class was selected based on evidence of TBDT domains (β-barrel, plug, loops, micelle). This 3D volume was used to generate templates, which were used in a template picker job on the entire 5960 micrograph dataset, resulting in ~7.7 m extracted particles (bin 2). These particles were plugged into a heterogeneous refinement using the 3D volume which was used for

template generation alongside the four "junk" classes. The best class was carried forward and used in Ab-Initio generation, with the top class again selected for heterogeneous refinement alongside the "junk" volumes. This process was repeated three times to remove as many junk particles as possible, then a final 2D classification was performed on the best remaining class to leave 304,879 particles, which were then re-extracted at box size $256 \times 256$. These particles and volume were added to a non-uniform refinement, and the resulting volume had its hand flipped by homogenous reconstruction. The particles were submitted to global CTF refinement, and then another non-uniform refinement using the hand-flipped volume. The particles from this volume were then rebalanced to ensure equal representation of viewing directions, and the remaining 102,419 particles were submitted to local refinement using a tight-fitting mask to omit the micelle, yielding a 2.9 Å final density map. A schematic representation of this workflow is shown in Supplementary Fig. 2.

An initial predicted model for Cir/MccV was generated in AlphaFold2[41] (version 2.3.2) using the multimer function, which was then docked into the EM density map using PHENIX[76]. The placed model was then manually adjusted in Coot[77], using a DeepEMhancer[78] map to aid visualization of density fit and rotamer assignments; local real-space refinement and regularization were performed against the experimental map. Final refinement and validation of the model was performed in PHENIX with additional relaxing in ISOLDE[79].

## Biolayer interferometry
BLI experiments were performed using a Gator Plus automated BLI instrument (Gator Bio). Purified MccV (WT and mutants) was biotinylated using an EZ-Link Sulfo-NHS-LC biotinylation kit (ThermoFisher) according to the manufacturer's protocol. Prior to experiments, streptavidin-coated SA-XT sensors (Gator Bio) were soaked in 300 µL PST-T in a 96-well MAX plate. In the instrument, sensors were equilibrated to baseline in PBS-T for 60 s, then MccV-biotin was loaded at a concentration of 100 nM. Loading time was 120 s. Loaded sensors were allowed to return to baseline for 120 s before the association phase. For association, Cir (WT and mutants) was serial twofold diluted in PBS-T (5 µM -> 78 nM), and these dilutions were added to the sensors for an association time of 300 s. After association, sensors were returned to PBS-T for 240 s of dissociation time. For experiments containing mutated Cir or MccV, a probe loaded with WT MccV and addition of 5 µM WT Cir was added as a positive control. Addition of PBS-T alone was used as a negative control. Kinetic analysis was performed in the Gator Navigator Software as follows: The Y-axis for sensorgrams was aligned to the beginning of the association phase with inter-step correction between association and dissociation included in alignment. The first 30 s of association and dissociation data were fit to a 1:1 global fit model with unlinked sensor Rmax, and $K_D$ was determined from the kon and koff values returned by the model fitting. An additional steady state equilibrium $K_D$ was calculated for WT Cir and MccV. Raw response data were plotted in GraphPad Prism for visualization.

## Competitive binding ELISAs
Two hundred microliters of purified, His-tagged Cir or Cir/Colicin Ia R Domain complex was seeded into the wells of a nickel-coated ELISA plate (ThermoFisher) at a concentration of 1 µM and allowed to adsorb for 1 h at room temperature. Separately, a third well was seeded with blocker (5% BSA (w/v) in PBS-T) alone. After seeding, liquid was aspirated and 300 µL blocker was added to all wells for 1 h at room temperature. Liquid was again aspirated before addition of 200 µL 1 µM biotinylated MccV was added as a probe. Probing proceeded for 1 h at room temperature, then liquid was aspirated and wells were washed three times with 300 µL PBS-T. Following washes, 200 µL of streptavidin-HRP (ThermoFisher, 1:5000 dilution in blocker) was added for 1 h at room temperature. Liquid was aspirated and wells were washed as described above before signal was developed by addition of 200 µL TMB substrate (ThermoFisher). Coloration proceeded for 3 min before the reaction was stopped by addition of 200 µL 0.5 N NaCl. Signal at 450 nm was recorded using a CLARIOstar Plus microplate reader

(BMG Labtech). Statistical analysis was performed in GraphPad Prism and significance was determined using one-way ANOVA with a 95% confidence interval.

## Sedimentation velocity analytical ultracentrifugation
NAPol solubilized Cir and its mixtures with MccV were analyzed by SV-AUC in phosphate-buffered saline. Sedimentation velocity experiments were conducted at 50,000 rpm (198,800 x $g$ at 7.1 cm) and 20 °C on a Beckman Coulter ProteomeLab XL-I analytical ultracentrifuge and An-50 Ti rotor. Sedimentation data were time-corrected and analyzed in SEDFIT[80] in terms of a continuous $c(s)$ distribution of Lamm equation solutions. Solution densities $\rho$, solution viscosities $\eta$, and protein partial specific volumes were calculated in SEDNTERP[81]. The protein refractive index increment was calculated in SEDFIT. The partial specific volume for NAPol was calculated based on its chemical composition following the method of Durchschlag and Zipper[82], and a refractive index increment of $0.149 \, \mathrm{cm}^3 \, \mathrm{g}^{-1}$ was used. Absorbance and interference $c(s)$ distributions were analyzed simultaneously using the fitted $f/f_o$ membrane protein calculation module in GUSSI[83] to obtain the protein and amphipol contributions to the sedimenting complex of interest.

## Flow cytometry
MccV RBD with an N-terminal dansyl modification was synthesized by GenScript. Strains were grown in LB media with appropriate antibiotics at 37 °C to exponential phase (0.2% arabinose was added to cultures containing pBAD plasmids encoding *cirA*). Cells were pelleted (4000 RPM, 10 min), supernatants were removed, and cells were gently resuspended in 1× PBS containing 50 mM glucose to a final cell density of $OD_{600} = 0.2$. For each condition tested, 1 mL of cell suspension was added to a 5 mL polystyrene round-bottom tube (Corning). For cells treated with MccV RBD-dansyl, resuspended peptides were added to a final concentration of 53 µM. Tubes were incubated in the dark for 30 min, then analyzed using a Sony SA3800 spectral cell analyzer until 10,000 events were captured. Results were analyzed, gated, and visualized using FlowJo software (v 10.10.0). The gating strategy for flow cytometry experiments is depicted in Supplementary Fig. 10.

## MccV bactericidal assays
The WT *cirA* sequence with an appended C-terminal V5 tag was cloned into a pBAD vector with ampicillin resistance using standard cloning techniques. WT MccV and its associated immunity protein sequence were previously cloned into a pBAD plasmid with kanamycin resistance. All Cir and MccV residue mutants were constructed using QuikChange Site-Directed Mutagenesis kit (Agilent). Cir mutant plasmids were used to transform a *cirA*::kan strain from the Keio collection in order to isolate the sensitivity of the Cir variant to MccV. Plasmids encoding MccV were used to transform an *E. coli* strain containing the microcin secretion system on plasmid pACYC[84].

Bactericidal activity was assayed using zone of inhibition assays. All strains were grown overnight at 37 °C in LB media with appropriate antibiotics. Prey strain cultures were pelleted (4000 RPM, 10 min), supernatants were removed, and cells were resuspended in LB media. Each prey strain was added to 10 mL of LB containing 0.75% agar and 0.2% arabinose to a final $OD_{600}$ of 0.001. The mixture was poured over a solidified bottom layer of LB containing 1.5% agar. Overnight cultures of bacteria containing the microcin secretion system and MccV plasmids were normalized to the same $OD_{600}$ in 500 µL, then pelleted (4000 RPM, 10 min). Cell pellets were resuspended in 50 µL of LB and 5 µL of predator strain was spotted onto plates containing the prey strains. Plates were grown overnight at 37 °C and zones of inhibition were inspected to infer mutant sensitivity.

## Expression test for Cir mutants
To ensure that variability in prey mutant sensitivity was not due to differences in protein expression, Cir production was assayed by western blot. Strains were grown overnight at 37 °C in LB with appropriate antibiotics and 0.2% arabinose. The following day, 1.5 mL of overnight culture was pelleted

and resuspended in 1 mL of 10 mM Tris pH 7.3 and 150 mM NaCl. All cultures were then normalized to the same density and 20 µL of cell suspension was mixed with SDS loading buffer and incubated at 99 °C for 10 min. Samples were loaded into a NuPAGE Bis-Tris 4–12% gel (Invitrogen) and run in MES buffer. Proteins were then transferred onto a PVDF transfer membrane with 0.45 µm pore size (Thermo Scientific), according to the manufacturer's instructions. After transfer, membranes were incubated in blocking buffer (1× TBS, 0.1% Tween, 5% milk) for 1 h at room temperature, then incubated overnight at 4 °C in blocking buffer containing anti-V5 tag antibody (1:5000 ratio). The membrane was washed three times in blocking buffer (10 min each) and incubated with blocking buffer containing anti-rabbit-HRP antibodies (1:1500 ratio) for 4 h at 4 °C. After incubation, the membrane was washed three times in TBST (1× TBS, 0.1% Tween). Membranes were briefly incubated with SuperSignal West Femto Maximum Sensitivity Substrate (Thermo Scientific) and imaged on a BioRad ChemiDoc imaging system.

## Reporting summary

Further information on research design is available in the Nature Portfolio Reporting Summary linked to this article.

## Data availability

The atomic coordinates of the Cir/MccV complex have been deposited at the Protein Data Bank under accession code 9NN6, and the accompanying cryo-EM 3D map has been deposited at the Electron Microscopy Data Bank under accession code EMD-49565. Data supporting these findings are available upon reasonable request. The numerical source data for the graphs in this paper can be found in the provided Supplementary Data file, and images of uncropped protein gels and western blots can be found in Supplementary Fig. 9.

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

## Acknowledgements

This study utilized resources from the NIH Multi-Institute Cryo-EM Facility (MICEF), the National Heart, Lung, and Blood Institute (NHLBI) Biophysics Core, and the NIH HPC Biowulf cluster (http://hpc.nih.gov). The authors would like to thank Yanxiang Cui, Huaibin Wang, Ulrich Baxa, and Bertram Canagarajah for microscopy technical support and data management, and Grzegorz Piszczek and Di Wu for biophysics core support. S.M., I.B., R.G., and S.B. are supported by the Intramural Research Program of the National Institute of Diabetes and Digestive and Kidney Diseases. A.O. and B.D. are supported by the National Institutes of Health (R01 AI182365, R56 AI179799), Welch Foundation F-2137, and Army Research Office W911NF2010195. This research was supported by the Intramural Research Program of the National Institute of Diabetes and Digestive and Kidney Diseases (NIDDK) within the National Institutes of Health (NIH). The contributions of the NIH author(s) are considered Works of the United States Government. The findings and conclusions presented in this paper are those of the author(s) and do not necessarily reflect the views of the NIH or the U.S. Department of Health and Human Services.

## Author contributions

S.M., A.O., B.D. and S.B. conceived the study. S.M. purified proteins, prepared cryo-EM grids, collected and analyzed cryo-EM data, built atomic models, designed mutants, performed binding analyses, and wrote the manuscript. A.O. designed mutants, performed flow cytometry, performed bactericidal assays, and wrote the manuscript. I.B. built atomic models and wrote the manuscript. R.G. collected and analyzed AUC data and wrote the manuscript. B.D. and S.B. directed the work, secured funding, and wrote the manuscript.

## Competing interests

The authors declare no competing interests.
