## [Transparent Peer Review file · Communications Biology]

Structural Insights into Cir-mediated Killing by the Antimicrobial Protein Microcin V

Corresponding Author: Dr Susan Buchanan

This manuscript has been previously submitted at another journal. This document only contains information relating to versions considered at Communications Biology.

Version 0:

Reviewer comments:

Reviewer #1

(Remarks to the Author)

Microcins, small antimicrobial peptides produced by a number of bacterial species to aid in inter-species competition, are to date largely uncharacterized structurally and biophysically. This manuscript by Maurakis et al. will do much to address this gap. Microcin V (MccV) binds to the TonB-Dependent Transporter (TBDT) Cir and utilizes TonB-dependent outer membrane active transport to traverse the outer membrane permeability barrier of Gram-negative bacteria. The cryoEM structure of the Cir/MccV complex presents clearly the detailed molecular interactions between Cir and MccV, and the similarity to colicin Ia binding to Cir is an interesting example of convergent molecular recognition. This structure identified likely-important Cir:MccV residue:residue interactions which were mutated and then evaluated for reduced MccV binding affinity - nicely confirming what was observed structurally. Lastly, the demonstration of MccV antimicrobial activity requiring functional TonB-dependent transport in the target bacteria is a pleasing confirmation. This work will be of interest to multiple scientific communities, including antimicrobial peptide researchers, those more widely interested in antimicrobials, membrane transport specialists and the more general membrane protein research community.

A number of minor suggestions and criticisms follow.

1. line 123. Provide a bit more description of MccV (microcin) immunity proteins and how they work, or at least one (or more) references.
2. line 143. What model did AlphaFold generate? The entire Cir:MccV complex or just MccV. I ask because y'all previously determined the Cir crystal structure so maybe you used that?
3. lines 214-232 (BLI to measure Kd). Please provide more information on how a Kd was obtained from this data. Consider including the equation(s) and software utilized. Also, why are lower-affinity Kd values for the mutants not reported vs. the semi-quantitative reporting of reduction of signal at a given probe concentration?
4. line 511. Please provide a reference or some other information regarding the microcin secretion system on that pACYC plasmid.
5. line 535 It was not clear from the Methods section how/where the V5 epitope was incorporated into Cir.

Reviewer #2

(Remarks to the Author)

The authors have resolved the structure of the MccV/Cir complex and used structure-guided mutagenesis to support their proposed mechanism. The data are significant and merit publication. The manuscript may be accepted after the authors address the following points:

1. The authors state that "MccV binds Cir similarly to Colicin Ia." Can they provide experimental evidence for competitive binding between MccV and Colicin Ia to Cir?
2. AlphaFold is known to have limitations in predicting intrinsically disordered regions. How well does the predicted position of MccV align with the cryo-EM density? Could the authors provide a commentary on this? Additionally, what is the local resolution at the binding interface in the complex? How can the authors be confident that the modeled structure shown in Figure 2 is accurate?
3. It would be clearer if the dissociation constant (KD) values were shown directly on the plots in Figures 5 and 6. Is there a

correlation between KD values and bactericidal activity, such as the minimum inhibitory concentration (MIC)? If the MIC is high, the potential for clinical application would be limited.

4. The description of Figure 1B needs clarification. Please revise for better understanding.

5. The axes in Figure 1C are unclear. Please label them more clearly.

6. In Figure 1D, what does "vector" refer to in the " Δ cirA + vector" group?

7. What is meant by "4-Met"? The Methods section mentions that the expressed Cir protein contains a 10 \times His tag and a TEV recognition site. However, there is no description of a TEV cleavage step. Please clarify.

8. How was the complex formed? Was it a 1:1 molar mixing? What buffer conditions were used?

9. Does E. coli BL21(DE3) express endogenous Cir?

Version 1:

Reviewer comments:

Reviewer #1

(Remarks to the Author)

This Reviewer is appreciative of the care and attention Maurakis et al. have devoted to the issues raised by the Reviewers. Except for one small request (see below), I consider Reviewer concerns to have been (more than) adequately addressed. This solid structure/function paper substantially increases our basic science knowledge of microcins, which could possibly have future translational benefit in development of novel antimicrobials.

==> The "one small request" is to please include the experimental uncertainties (i.e., the errors) in the Kd determinations depicted in Figure 5D and 6C. Thank you <==

Reviewer #2

(Remarks to the Author)

The authors did not answer whether there is a correlation between KD values and bactericidal activity (part of my comment 3)

Version 2:

Reviewer comments:

Reviewer #2

(Remarks to the Author)

The revision and explanation are satisfying. This manuscript is ready for publication.

The authors would like to thank the reviewers for their comments and suggestions regarding our manuscript *Structural Insights into Cir-mediated Killing by the Antimicrobial Protein Microcin V*. Several helpful questions and requests were made which we believe ultimately served to strengthen our report. Please find below our responses.

Reviewer #1:

1. line 123. Provide a bit more description of MccV (microcin) immunity proteins and how they work, or at least one (or more) references.

Much specific information about the *cvi* gene product is still not understood. Gilson *et al.* noted that the predicted protein contains a hydrophobic region from residues 10-32, which is hypothesized to interact with the inner membrane. If so, Cvi would bear similarities to the immunity proteins described for Colicins E1, A, Ia, and Ib. Additionally, *cvi* is iron-repressed (as is MccV secretion itself), consistent with a model wherein MccV-based competition is stimulated by the same low-iron conditions which induce Cir production in prey species.

Additional text and a few extra references have been added to the introduction section (lines 78-79) to support this.

2. line 143. What model did AlphaFold generate? The entire Cir:MccV complex or just MccV. I ask because y'all previously determined the Cir crystal structure so maybe you used that?

AlphaFold was used to generate a prediction for Cir in complex with MccV. As one might expect for AlphaFold, a considerable amount of the predicted structure for MccV was low confidence, but the residues corresponding to the RBD were reasonable and served as a sufficient starting point for us to continue modeling manually.

Please see below images of the AF2 model colored by b-factor (left) and chain (right).

3. lines 214-232 (BLI to measure Kd). Please provide more information on how a Kd was obtained from this data. Consider including the equation(s) and software utilized. Also, why are lower-affinity Kd values for the mutants not reported vs. the semi-quantitative reporting of reduction of signal at a given probe concentration?

We have updated the reporting for Kd measurements in response to this comment and comment #3 from Reviewer #2. Affinity is now reported for all Cir and MccV variants which were tested instead of the semi-quantitative comparison which was used before. Using the Gator Plus software (same software that was used for BLI measurements), we aligned the Y-axis for sensorgrams to the beginning of the association phase, including inter-step correction between association and dissociation, and fit the data to a 1:1 global fit model using unlinked sensor Rmax (each sensor's sensitivity is considered independently instead of averaging out) and included the first 30 seconds of association and dissociation in the fit. From this, Kd was calculated from the koff and kon rate constants instead of as steady state ($[A][B]/[AB]$) as had been done previously.

A table showing Kd and MccV sensitivity is now shown in Figures 5 and 6, and the methods section has been updated to include information on model fitting (lines 225-229 and 514-519 of revised text).

4. line 511. Please provide a reference or some other information regarding the microcin secretion system on that pACYC plasmid.

A reference has been added to the end of this section (<https://doi.org/10.1128/aem.00335-23>, reference #84 in the revised submission). Briefly, the MccV secretion complex is composed of CvaA, CvaB, and TolC. Our experimental secretion system uses a two-plasmid approach: *cvaAB* is expressed from pACYC, while *cvaC* (MccV) and *cvj* (immunity protein) are expressed from pBAD. The native *tolC* from the chromosome is used.

5. line 535 It was not clear from the Methods section how/where the V5 epitope was incorporated into Cir.

The V5 epitope tag was appended to the C-terminus of Cir using a QuikChange site-directed mutagenesis kit. This information can be found in the "MccV Bactericidal Assays" section of the methods beginning on line 502 of the original submission (line 567 of the revised submission).

Reviewer #2:

1. The authors state that "MccV binds Cir similarly to Colicin Ia." Can they provide experimental evidence for competitive binding between MccV and Colicin Ia to Cir?

We agree that inclusion of a competitive binding experiment is a strong addition to this report, and we thank the reviewer for this suggestion. To address this, we purified Cir in complex with the R Domain of Colicin Ia, and added either this complex or apo-Cir to 96-well plates for an ELISA-type analysis. We probed these receptors with purified, biotinylated MccV before a secondary probe with HRP-conjugated streptavidin to assess MccV binding. We found that

MccV did not bind Cir above background levels when the Colicin Ia R Domain was already present, consistent with our observation that the two ligands share the same binding site.

These data have been added as panels C-E of Supplementary Figure 4, and the text has been amended to include these observations (lines 197-203).

2. AlphaFold is known to have limitations in predicting intrinsically disordered regions. How well does the predicted position of MccV align with the cryo-EM density? Could the authors provide a commentary on this? Additionally, what is the local resolution at the binding interface in the complex? How can the authors be confident that the modeled structure shown in Figure 2 is accurate?

During model building, we discarded AF2 predicted residues that had very low confidence, which eliminated a large portion of the MccV prediction (see above the b-factor colored predicted model). However, the residues corresponding to the RBD were sufficiently close to the experimental EM map that we were able to manually adjust the predicted model to an appropriate fit. Side chains for residues Trp78, Tyr70, and (to a certain extent) Leu83 and Asp85 were of particular help during placement.

The local resolution of this section is shown in two images below. Overall, for MccV, the local resolution ranged from ~ 3 Å for the areas most buried within Cir, down to $3.7+$ Å for those at the periphery (top image). For the binding pocket specifically, we show the local resolution with the side chains of R436 and R490 highlighted, in which the resolution is ~ 2.8 - 3.0 Å (bottom).

Overall, the portion of the AF2 model corresponding to Cir showed high confidence, with lower confidence for MccV. See below the predicted model colored according to pLDDT, and the accompanying PAE matrix:

Very high (pLDDT > 90) Confident (90 > pLDDT > 70) Low (70 > pLDDT > 50) Very low (pLDDT < 50)

■ ■ ■ ■

We discarded residues with very low (<50) confidence and used those remaining for manual adjustment.

Map-model alignment of various sidechains in final model:

Collectively, between visual inspection of the model inset into the experimental map, the appropriate fit of prominent sidechains, the reported map-model correlation coefficient reported by PHENIX (0.88), and the local resolution of the map in the areas which constitute the binding pocket, we are confident that our model is an appropriate representation of the complex based on available data.

We have added the AF2 predicted model and PAE matrix, along with a legend for pLDDT scores, as Supplementary Figure 3 in the revised submission.

3. It would be clearer if the dissociation constant (KD) values were shown directly on the plots in Figures 5 and 6. Is there a correlation between KD values and bactericidal activity, such as the minimum inhibitory concentration (MIC)? If the MIC is high, the potential for clinical application would be limited.

We have updated the reporting for Kd measurements in response to this comment and comment #3 from Reviewer #1 above. Affinity is now reported for all Cir and MccV variants which were tested instead of the semi-quantitative comparison which was used before. Using the Gator Plus software (same software that was used for BLI measurements), we aligned the Y-axis for sensorgrams to the beginning of the association phase, including inter-step correction between association and dissociation, and fit the data to a 1:1 global fit model using unlinked sensor Rmax (each sensor's sensitivity is considered independently instead of averaging out) and included the first 30 seconds of association and dissociation in the fit. From this, Kd was calculated from the koff and kon rate constants instead of as steady state ($[A][B]/[AB]$) as had been done previously.

A table showing Kd and MccV sensitivity is now shown in Figures 5 and 6, and the methods section has been updated to include information on model fitting (lines 225-229 and 514-519 of revised text).

4. The description of Figure 1B needs clarification. Please revise for better understanding.

The figure legend for 1B has been expanded for added information and clarity.

5. The axes in Figure 1C are unclear. Please label them more clearly.

The graphs in 1C have been enlarged slightly and the axes and labels have been modified to hopefully improve clarity.

6. In Figure 1D, what does "vector" refer to in the " Δ cirA + vector" group?

This refers to a *cirA* knockout strain that has been transformed with an empty counterpart to the *cirA* complementation plasmid and therefore still does not produce Cir. The label for this image has been changed and additional text added to the figure legend to improve clarity.

7. What is meant by "4-Met"? The Methods section mentions that the expressed Cir protein

contains a 10×His tag and a TEV recognition site. However, there is no description of a TEV cleavage step. Please clarify.

The plasmid used for Cir purification in our study contains the same *cirA* gene that was used in the 2007 EMBO Journal paper describing the Cir/Colicin Ia interaction (<https://doi.org/10.1038/sj.emboj.7601693>). This gene had residues W338, L343, F589, and V591 mutated to methionine, and the crystal structure was solved using multiwavelength anomalous diffraction from a selenomethionine-derived crystal. Therefore, we have kept the “4-Met” nomenclature. These mutations are all located in the β -barrel and are secluded from the ligand binding pocket, so no impact on MccV binding should be expected. Furthermore, all functional experiments to assess MccV cytotoxicity were performed using strains that lack these mutations.

For clarity, additional text describing these mutations has been added to the methods section (lines 355-357).

8. How was the complex formed? Was it a 1:1 molar mixing? What buffer conditions were used?

To form the complex, we added an entire preparation of purified MccV to Cir while Cir was still in native *E. coli* membrane (prior to solubilization with DDM) and allowed them to mix for ~2h, then proceeded to solubilization and affinity capture of Cir. At this early stage in our study, we did not know the stoichiometry of the complex, so we instead opted to add MccV in excess. We have retrospectively looked at the concentrations of the two protein preps and determined that MccV was added at approximately a 2:1 molar ratio, with a 1:1 complex forming. The buffer at this stage was PBS pH 7.5.

Additional text has been added to the methods section (lines 426-428) to clarify this.

9. Does *E. coli* BL21(DE3) express endogenous Cir?

Yes. In fact, a WT and *cirA* isogenic mutant of BL21 (DE3) were used to demonstrate presence/absence of Colicin Ia sensitivity in doi.org/10.1038/sj.emboj.7601693. Because our method of Cir purification was nickel affinity, we do not expect that the endogenous Cir, which contains only 9 native histidine residues, contributed meaningful contamination to our purification. Furthermore, MccV was expressed as an inactive MBP fusion from BL21 (DE3), so Cir-dependent self-toxicity during this process was not a concern despite the lack of the Cvi immunity protein.

The authors thank the reviewers for their follow-up review of our manuscript *Structural Insights into Cir-mediated Killing by the Antimicrobial Protein Microcin V*. We have attempted to address the two remaining issues as described below:

Reviewer #1:

1. The "one small request" is to please include the experimental uncertainties (i.e., the errors) in the K_d determinations depicted in Figure 5D and 6C.

Experimental uncertainties have been added to the charts in Figure 5D and 6C. Also fixed a typo-related mistake for C435A.

Reviewer #2:

1. The authors did not answer whether there is a correlation between K_D values and bactericidal activity (part of my comment 3)

We have attempted to address this in the text added at lines 323-336. Because our killing assay utilizes naturally exported MccV and not purified biomolecules, a true calculation of MIC was not included in this study. Our data suggest that reduction in binding by the Cir and MccV mutants is consistent with loss/reduction in bactericidal activity, but the relationship is not directly correlated (as demonstrated by D85A MccV having lower affinity but better activity than D85R). This may be a result of changes to import which arise from mutagenesis but are not directly assessed by BLI, or potentially from differences in how well D85A and D85R affect SdaC after import. We anticipate future studies building off these initial findings to address these uncertainties directly.